# Research on the Properties of Zein, Soy Protein Isolate, and Wheat Gluten Protein-Based Films Containing Cellulose Nanocrystals

**DOI:** 10.3390/foods11193010

**Published:** 2022-09-27

**Authors:** Menghan Fu, Mengyuan Cao, Jiangkai Duan, Qin Zhou, Mengxue Dong, Ting Zhang, Xuebo Liu, Xiang Duan

**Affiliations:** 1College of Food Science and Engineering, Northwest A&F University, Xianyang 712100, China; 2Jilin Provincial Key Laboratory of Nutrition and Functional Food, College of Food Science and Engineering, Jilin University, Changchun 130062, China

**Keywords:** zein, soy protein isolate, wheat gluten protein, cellulose nanocrystals, physicochemical properties

## Abstract

Plant protein films are a research hotpot in the current food packaging field for their renewable and bio-compatibility, and further improving the physicochemical properties of plant protein films in combination with biodegradable materials is of great significance. In this study, we selected cellulose nanocrystals (CNC) to modify the protein films with soybean protein isolate (SPI), wheat gluten protein (WGP), and Zein, and the physicochemical properties were studied. The results showed that the hardness and opacity of Zein-based films decreased by 16.61% and 54.12% with the incorporation of CNC, respectively. The SPI-based films performed with lower hardness and higher tensile strength. The thickness and opacity of WGP-based films increased by 39.76% and 214.38% after combination with CNC, respectively. Accordingly, this study showed that CNC could largely modify the physicochemical properties of the plant protein films, which provided a reference for the preparation of modified plant protein films using biodegradable materials.

## 1. Introduction

At present, the primary raw material of daily plastic products is petroleum. However, petroleum is a non-renewable resource, and petroleum-based plastics are difficult to degrade in nature and can cause damage to the ecosystem of nature [1]. According to estimates, 8.3 billion metric tons of plastics have been produced since its invention, out of which 40% have been used for packaging. According to statistics, global plastic waste is expected to reach 1.20 × 10^10^ t by 2050 [2]. To solve the problem, the most novelty and eco-friendly way are to use biodegradable plastics, such as degradable materials made from protein, cellulose, and other resources [3]. Among these resources, plant proteins are safe and low-cost [4], and plant protein films have been considered one alternative due to their remarkable barrier properties against ultraviolet (UV) light and non-polar substances.

Zein is an alcohol soluble protein in maize endosperm, accounting for about 60% of the total zein [5]. More than half of the amino acids of Zein are hydrophobic, making it one of the few natural proteins soluble in a hydrated organic solvent such as ethanol but not in water. Compared with other protein films, raw Zein films have shown unique features in the aspect of low permeability, glossy appearance, radiation-resistant and biodegradability [6], but the Zein film is unfit for food packaging application due to its poor mechanical properties. Vahedikia et al. [7] incorporated cinnamon essential oil (CEO) and chitosan nanoparticles into Zein-based films, respectively, causing a drastic decrease in water vapor permeability and an increase in tensile strength. Chen et al. [8] modified Zein-based films by a two-step method consisting of chitosan, followed by exposure to cold plasma, which increased their elongation at break.

Soy protein isolate (SPI), as a by-product in the process of grain and oil industrial production, is a highly polished and purified form of soy protein with a protein content higher than 90% [9]. SPI is mainly composed of albumin and globulin and has suitable film-forming properties. SPI is a popular biodegradable film material due to its wide practicability, low-cost, and eximious film-forming properties [10]. However, the characters such as poor mechanical properties, high water solubility, and hydrophilicity have restricted their application in food packaging. Liu et al. [11] studied that the addition of sodium caseinate (SC) prominently enhanced the tensile strength and transparency of SPI-based films, and transglutaminase (TGase) treatment significantly improved the tensile strength of SPI-based films.

Wheat gluten protein (WGP) is a by-product of wheat processing. It is composed of high-molecular-weight glutenin and low-molecular-weight gliadin, with a ratio of about 1:1 [12]. Glutenins, composed of polypeptide bonds polymerized by intermolecular disulfide bonds, are insoluble in water or common solvents and are responsible for the strength and stability of glutens [13]. In comparison, gliadins are soluble in aqueous ethanol and are responsible for their elasticity [14]. Glutenin and gliadin endow WGP films with good viscoelasticity, but WGP films have poor water resistance and hardness. Nataraj et al. [15] developed films from banana fibers and WG by solution casting and later compression molding in order to improve mechanical properties and resistance to moisture.

Cellulose nanocrystals (CNC) are cellulose-based nanoparticles that were extracted from natural source materials by acid hydrolysis [16]. CNC has been rated as one of the top 20 new materials with the most potential in the future due to its biodegradable, non-toxic and sustainable, having a broad application prospect in the field of compatible materials [17]. Some current studies have shown that CNC incorporated into films contributed to improving the mechanical and barrier properties of biocomposites [18]. Xiao et al. [19] studied that the addition of CNC could reduce the porosity and increase the penetration path of gas molecules inside the composite films, as well as enhance the mechanical strength of the composite films. 

Plant protein films can be used for food packaging, thereby extending the shelf life of food and ensuring the quality of food during transportation [20]. The protein films with better barrier performance can be used to package candy, fruit, and other hygroscopic foods, and protein films with high opacity can be used to package food that is away from light [21]. However, due to the limitation of single protein film material at present, modified materials such as plasticizers and antioxidants should be added to improve the mechanical, optical, and barrier performance of protein films [22]. 

Therefore, the objective of the present research was to evaluate the effect of CNC on the physical properties, water resistance, and mechanical properties of the three protein films (Zein, SPI, and WGP), which provided a theoretical reference for the research and preparation of biodegradable materials.

## 2. Materials and Methods

### 2.1. Materials 

Zein and SPI were purchased from Shanghai Yuanye Bio-Technology Co, Ltd. (Shanghai, China). WGP was purchased from Midaner Trading Co. (Henan, China). CNC and glycerin were provided by Guanghua Sci-Tech Co., Ltd. (Guangdong, China). Sodium hydroxide and hydrochloric acid were supplied by Tianjin Damao Chemical Reagent Factory. All other chemical reagents were analytical grade.

### 2.2. Film Preparation

The process of film preparation is illustrated in Figure 1.

Zein films: 10.0 g of Zein and 0%, 2.5%, 5% (*w/w*) CNC were dispersed in 100 mL 75% ethanol solution, respectively. The solutions were stirred at room temperature for 20 min, followed by the addition of 50% glycerin (*w/w*). The solutions were labeled Zein (0% CNC), Zein-CNC-1 (2.5% CNC) and Zein-CNC-2 (5% CNC). The solutions were stirred at 70 °C for 30 min and cooled down to room temperature. The solutions (6 mL) were cast in circular Teflon molds with a diameter of 9 cm, followed by steam drying in a water bath at 70 °C for 2 h.

SPI films: 6.0 g of SPI and 0%, 2.5%, 5% (*w/w*) CNC were dispersed in 100 mL deionized water, respectively. The solutions were stirred at room temperature for 20 min and adjusted to pH 10.0 with 1 M NaOH, followed by the addition of 50% glycerin (*w/w*). The solutions were labeled SPI (0% CNC), SPI-CNC-1 (2.5% CNC), and SPI-CNC-2 (5% CNC). The solutions were stirred at 70 °C for 30 min and cooled down to room temperature. The solutions (10 mL) were cast in circular Teflon molds with a diameter of 9 cm, followed by drying in a vacuum oven at 50 °C for 5 h. 

WGP films: 6.0 g of WGP and 0%, 2.5%, 5% (*w/w*) CNC were dispersed in 100 mL 60% ethanol solution, respectively. The solutions were stirred at 60 °C for 20 min and adjusted to pH 11.0 with 1 M NaOH, followed by the addition of 50% glycerin (*w/w*). The solutions were labeled WGP (0% CNC), WGP-CNC-1 (2.5% CNC), and WGP-CNC-2 (5% CNC). The solutions were stirred at 70 ℃ for 20 min and cooled down to room temperature. The solutions (10 mL) were cast in circular Teflon molds with a diameter of 9 cm, followed by drying in a vacuum oven at 50 °C for 5 h. 

Finally, all films were preconditioned in a constant temperature and humidity chamber (25 °C, 50% relative humidity) for at least 48 h to normalize the moisture content before further experiments.

### 2.3. Physical Appearance

#### 2.3.1. Thickness

The thickness of the film was measured with a micrometer. The thickness of film samples was determined from an average of three random measurements. 

#### 2.3.2. Color

A light color measurement instrument (CS-820, Caipu, China) was used to measure the film color. The color differences (ΔE*, L*, a*, and b* value) could be directly read from the instrument.

#### 2.3.3. Opacity

The film samples were cut into 30 × 10 mm pieces, and the thickness of each piece was measured by a micrometer. The absorbance of the film at 600 nm was measured by an ultraviolet spectrophotometer (UV2550, SHIMADZU, Japan). The opacity value of the film was expressed as the ratio between the absorbance at 600 nm and the film thickness (mm) [23].

### 2.4. Mechanical Performance

#### 2.4.1. Tensile Test

The tensile test of the film was measured on a texture analyzer (TA. XT Plus, StableMicro Systems Ltd., UK). The film was cut into a rectangle of 1 cm wide. The initial distance was set to 10 mm, the stretch speed was 1 mm·s-1, and the tensile force was 10 g [24]. The tensile distance was set to 60 mm. The tensile strength of the calculation followed the Equation (1):Tensile strength = F × g/(w × d)(1)

F is the breaking force (kg), g is the local acceleration of gravity (m·s^−2^), w is the width of the film (m), and d (m) is the thickness of the film.

#### 2.4.2. Puncture Test

Each film was mounted on a compression device with a 10 mm round hole and perforated by a P/2 cylindrical probe moving at 0.1 mm·s^−1^. Force-deformation curves were obtained, and force (N) at the puncture point was then recorded to represent the hardness (N) of the films.

### 2.5. Water Resistance Performance

#### 2.5.1. Moisture Content

Each film was placed in a clean petri dish. Then, it was dried at 30 °C until a constant weight was obtained. The moisture content was calculated according to the following Equation (2): Moisture content (%) = (m_1_ − m_2_)/m_1_ × 100%(2)
where m_1_ and m_2_ are the initial weight (g) and final dry weight (g) of film samples, respectively.

#### 2.5.2. Total Soluble Matter (TSM)

The film, after the moisture content test was placed in a sealed tube containing an appropriate amount of deionized water, and sealed tubes were stirred at room temperature for 24 h. After that, the film was dried at 50 °C until a constant weight was obtained. The percentage of weight loss is defined as the TSM value. TSM was calculated as Equation (3):TSM (%) = (m_1_ − m_3_)/m_1_ × 100%(3)
where m_1_ and m_3_ are the initial dry weight (g) and final dry weight (g) of film samples, respectively.

#### 2.5.3. Water Vapor Permeability (WVP)

Glass jars (10 mL) were filled at the bottom with 40 g of anhydrous calcium chloride (0% relative humidity) and sealed with the film. Then, the jars covered with films were placed in a desiccator and weighed after a week. WVP was calculated with the following Equation (4):WVP = Δm × d × 24/(A × T × ΔP)(4)
where ∆m (g) refers to the quality difference of the wide-mouth bottle; d (mm) is the film thickness; A (m^2^) is the area of the film; ∆P (kPa) refers to the WVP difference on both sides of the film; T (h) is the measurement time.

#### 2.5.4. Water Contact Angle (WCA)

The WCA of the film was measured by the water contact angle measuring instrument (JY-PHb, Jinghe, China). Water droplets (2 μL) were dropped onto the surface of the film. The value of the water contact angle could be directly obtained by drawing a diagram with software [25].

### 2.6. Scanning Electron Microscope (SEM)

A scanning electron microscope (SEM, HITACHI, Japan) was used to observe the surface and cross-section morphology of the film. The surface was magnified 1500 times, and the cross-section was magnified 550 times [26].

### 2.7. Data Analysis

All of the tests were carried out three times, and the parallel mean value was reported with standard deviation. The Origin Lab software was utilized for statistical analysis. Comparisons were carried out by using the Tukey test analysis in ANOVA. The significant differences were based on a 95% confidence level.

## 3. Results and Discussion

### 3.1. Physical Appearance

#### 3.1.1. Thickness

The thicknesses data of films are listed in Table 1, Table 2 and Table 3, all film samples had thicknesses in the range of 86.67–163.00 μm. The thickness of the composite films depends not only on the total amount of non-solvent in the film-forming solution but also on the intermolecular interactions in the composite films. Among the three types of raw protein films, the thickness of SPI film was the lowest (104.33 μm), and the Zein film was the highest (126.67 μm). The thickness of the three types of protein-CNC films showed different trends with the increasing concentration of the CNC. The results indicated that the incorporation of CNC into Zein film increased the thickness from 126.67 μm (Zein) to 163.00 μm (Zein-CNC-1), followed by reducing it to 86.67 μm (Zein-CNC-2). When the amount of CNC was 2.5%, the composite structure was relatively loose, but when the CNC increased from 2.5% to 5%, it formed a denser structure than the raw Zein film. The thickness of SPI film decreased from 104.33 μm (SPI) to 99.33 μm (SPI-CNC-2) when CNC was incorporated. The interactions between CNC and SPI reduced the distance between the micelles to form a relatively stable and dense network structure [27]. The thickness of WGP film increased from 109.00 μm (WGP) to 154.67 μm (WGP-CNC-2). CNC reacted with WGP particles, and they were cross-linked by a disulfide bond [28], altering the compact structure of the WGP film and leading to a looser structure. This result was in line with the research processed by Sukyai et al. [29], indicating that the thickness of whey protein isolate-based film increased after incorporating CNC from sugarcane bagasse.

#### 3.1.2. Color

The color data of the films are shown in Table 1, Table 2 and Table 3. The L* value represents light and dark, the a* value represents red and green, the b* value represents yellow and blue, and the ΔE* value represents the chromatic aberration change. With the increase of CNC, the color changes of the three types of protein films were different. When CNC was 2.5%, the L* value of WGP film decreased, which may be due to the aggravation of the Maillard reaction, resulting in the darker color of the WGP film [30]. After adding CNC, all the a* values (−represents green) decreased compared with raw protein films, respectively. The b* values (+represents yellow) of Zein film and SPI film didn’t significantly change, but the b* value of WGP film showed an upward trend. The ΔE* value of the WGP film and SPI film exhibited an increasing trend when CNC increased to 2.5%. The increase in ΔE* was probably because of the inner color of the CNC [31]. 

#### 3.1.3. Opacity

The opacity result of film samples is shown in Figure 2. The opacity of Zein-based films was higher than that of SPI-based films and WGP-based films. This may be due to the high amount of zeaxanthin in Zein, which contributed to the yellow appearance of the Zein-based films. The appearance of SPI and WGP were white or milky white, so the opacity of SPI-based films and WGP-based films was lower. The opacity of the Zein-CNC films was lower than that of the Zein film without CNC. There were studies reporting that CNC transferred from the substrate of the films to its surface in the process of films drying and forming [7]. This phenomenon caused the decrease in the yellow color of the protein films, which indirectly led to the decline in the opacity of the protein films. As shown in Figure 2, the opacity of the WGP film and the SPI film significantly enhanced after adding CNC. This was because CNC dispersed in the protein films and thus scattered or blocked the light [32]. As the amount of CNC increased, the light barrier properties gradually enhanced, resulting in a significant increase in the opacity of protein films. This trend was in agreement with the recent research showing that the absorbance of the κ-carrageenan/CNC nanocomposite films increased with increasing fiber loading [33]. 

### 3.2. Mechanical Properties

#### 3.2.1. Tensile Test

The tensile test results are shown in Figure 3. The tensile strength reflects the fracture resistance of films. It could be seen that the tensile strength of Zein film was the highest (9.02 MPa), and WGP film was the lowest (0.57 MPa). 

The effects of CNC on the Zein-based films’ tensile strength were different from the other two protein-based films. The Zein film’s tensile strength reduced when it was incorporated with CNC, which was because the weak network formed under the effect of CNC effectively changed the toughness and reduced the brittleness of the Zein film. The incorporation of CNC improved the tensile strength of SPI film. The cross-link with CNC changed the mechanical properties and flexibility of the SPI particles [34]. This could be explained by the interaction between the carboxyl group and the amino groups of SPI, and the hydroxyl groups of CNC via hydrogen bonds in the process of film-forming. However, the tensile strength of the SPI-CNC-2 film was lower than that of the SPI-CNC-1 film, which was because the agglomeration of CNC caused the structure of SPI-CNC films to defect [35]. These results showed that the proper amount of CNC could enhance the strength of SPI films. When the addition amount of CNC was 2.5%, the tensile strength of WGP films was the highest. Previous literature reported that the stronger interfacial interaction appeared through chemical bonds between the CNC matrix and the WGP interfacial area [36]. The tensile strength of the WGP-CNC-2 film was lower than that of the WGP-CNC-1 film, which can be explained by the fact that excessive CNC can’t be adequately dispersed in the WGP-CNC-2 film and loosen the structure of the WGP-CNC-2 film. 

#### 3.2.2. Puncture Test

The hardness results are shown in Figure 3. The Zein film possessed a higher hardness than that of the other two protein films, which was mainly because of the strong intermolecular interaction and the close molecular connection between Zein molecules [37]. The hardness of the Zein, SPI, and WGP films were all decreased when CNC was added. It might be because CNC had a compact structure and formed a percolating network within the polymer matrix, thus changing the hardness of the films [38]. Meanwhile, the filling effect of CNC and the strong interfacial interaction between CNC and film matrix would also reduce the hardness of films. This suggested that it is more suitable to improve the mechanical properties and reduce the hardness of the films by adding CNC to make the film more convenient for processing for specific applications. This result was in agreement with previous studies showing that the addition of CNC could enhance the mechanical properties of protein films [32]. Additionally, the hardness of WGP film was lower than that of the other two protein films, which could be explained by the elastic behavior of glutenins and the strong viscoelasticity and extensibility of WGP.

### 3.3. Water Resistance Performance

#### 3.3.1. Moisture Content, Water Vapor Permeability, and Total Soluble Matter 

The data on moisture content, water vapor permeability (WVP), and total soluble matter (TSM) of protein films are shown in Table 4, Table 5 and Table 6 and Figure 4.

As shown in Table 4, there were no statistical differences in moisture content among the Zein film, Zein-CNC-1 film, and Zein-CNC-2 film. As shown in Figure 4, the TSM value of the Zein-CNC-1 film (84.42%) was significantly higher than that of Zein film (52.70%), which was due to the more hydrophilic groups of CNC. However, the Zein-CNC-2 film exhibited lower TSM values than that of the Zein-CNC-1 film. We found a similar result in the cellulose nanofibrils-biopolymer nanocomposite films [37], which could be explained by a relatively compact structure formed of CNC and Zein impairing the ability of films to absorb water. The Zein-CNC-2 film exhibited lower WVP values than that of the Zein-CNC-1 film. This might be because an excess amount of CNC resulted in a filling effect and a more compact film structure of hydrophobic interaction, leading to reduced porosity [39].

For SPI films, there was no statistical difference in the moisture content and WVP values between the SPI films with and without the addition of CNC (Table 5). Our results indicated that the incorporation of CNC into the SPI matrix decreased the TSM value firstly decreased and then began to increase (Figure 4). Some relative research studies reported the same results, which was explained by the hydrogen bond interaction and physical cross-linking structure between CNC and SPI [40,41]. This was because a tighter structure of the film formed by the hydrogen bond interaction between CNC and SPI, which reduced the contact between hydrophilic groups and water molecules, resulting in lower solubility [42]. Xiao et al. [19] reported a similar result that the TSM value of the nanocomposite films was significantly reduced as a result of reinforcement incorporation, particularly in the CNC-containing agents. For the SPI-CNC-2 film, the TSM value was significantly higher than that of the SPI-CNC-1 film. It has been reported that excessive addition of CNC was prone to promote agglomeration and could not be uniformly dispersed in the composite structure to form a tight structure with SPI, leading to the exposure of hydrophilic groups and a higher solubility [32,34]. 

For WGP films, the structure of WGP film was a heterogeneous spatial network structure composed of polypeptide chains connected by intermolecular disulfide bonds [43]. There was no significant difference in the moisture content and TSM values between the films with or without CNC (Table 6 and Figure 4), indicating that the addition of CNC exerted no significant influence on the water resistance of WGP film. The WGP-CNC-1 film exhibited a higher WVP value than that of the WGP and WGP-CNC-2 film. According to previous literature, this might be because CNC destroyed the structure of the film, leading to the change in the structure of films, and thus, the porosity of the film increased [34]. The lower WVP value of the WGP-CNC-2 film went down for the same reason as the Zein-CNC-2 film.

#### 3.3.2. Water Contact Angle (WCA)

The WCA is an indicator of the hydrophobicity of the surface of the film. The films with WCA that exceeds 65° can be considered hydrophobic surfaces (Figure 5). The WCA of SPI film (48.13°) was significantly higher than that of WGP film (29.24°). Meanwhile, the results showed that the WCA of the SPI-CNC-1 film (40.39°) and the SPI-CNC-2 film (45.45°) was lower than SPI film, indicating that the hydrophobicity of the surface of SPI film decreased after adding CNC. CNC contained a large number of hydrophilic groups, and SPI possessed both hydrophilic groups and hydrophobic groups [44,45]. It has been reported that the SPI film incorporated into CNC formed a structure of hydrophobic groups inside and hydrophilic groups bonded by hydrogen bonds outside [46]. Xiao et al. [19] reported similar results, suggesting that the CNC were rich in hydroxyl groups to bind water molecules, reflected by the gradually decreasing WCA values. The WCA of the SPI-CNC-2 film was higher than that of the SPI-CNC-1 film, which was due to the agglomeration of excessive CNC that destroyed the original structure and led to rough surfaces. Wesołowska et al. [47] showed that the biopolymer surface is hydrophilic, and the rough structure of the film could increase the WCA value. Similarly, the SPI-CNC-2 film had a rougher surface than that of the SPI film and the SPI-CNC-1 film (Figure 6). The WGP film was a three-dimensional network structure mainly formed by the interaction of hydrophobic bonds and disulfide bonds. It is notable that no accurate data were available for Zein films due to a large number of cavities and voids in the loose structure that allowed water droplets to rapidly penetrate the films and could not be measured [48].

### 3.4. SEM

The physicochemical and processing properties of films are primarily dependent on the regularity of the molecular arrangement of macromolecules in the process of film formation. According to Figure 6, some irregular spots appeared on the Zein film and SPI film, but the WGP film exhibited a relatively smooth and homogeneous surface. Moreover, some irregular folds appeared on Zein film but were not observed in other films.

**Figure 6 foods-11-03010-f006:**
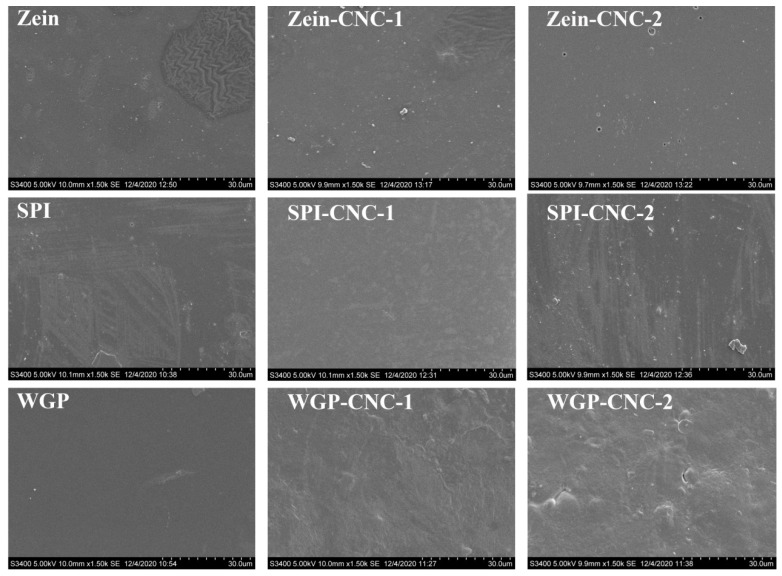
SEM images of rough surface of the Zein/SPI/WGP-based films (magnifications are 1000×).

CNC played an essential role in the compatibility of the films [49]. With the increasing amount of CNC, there was no noticeable change on the surface of the Zein-CNC films, indicating that the compatibility of CNC and Zein was excellent. Compared with the SPI film, the irregular white spots on the surface of the SPI-CNC-1 film were reduced due to a more compact structure formed by CNC and SPI, which can also be reflected in the tensile test and TSM value results. Similarly, the internal structure of the SPI-CNC-1 film surface was denser than that of the raw SPI film (Figure 7). Wu et al. [50] studied the morphology of the SPI films with different mixing ratios of CNC, showing that SPI-CNC films exhibited a fine continuous structure. There were some small particles that appeared on the surface of the SPI-CNC-2 film, which led to the heterogeneity of the film. This might be because that massive CNC was dissociated and prone to agglomerate, which was reflected in the TSM value and WCA results. With the increasing amount of CNC, the surface of WGP films appeared to have more bubble-like bulges. This was because of the phase separation and the destruction of the network structure of WGP due to the addition of CNC, which was also reflected in the thickness and WCA results. 

## 4. Conclusions

In this study, a series of composite protein films with three kinds of plant proteins and CNC were prepared. The addition of CNC improved the physical appearance, mechanical properties, and water resistance performance of three types of protein films. The results demonstrated that the addition of CNC significantly reduced the hardness of the protein films, which made them more convenient to be processed and produced. The addition of CNC had a significant impact on the color and opacity of the protein films and other physical properties. This can enable protein films with different properties to be used in the packaging of different foods and expand the application range of protein films. However, CNC has no significant effect on improving the water resistance of protein films, which still needs further research and discussion in the future. This study provides a theoretical basis for the design and development of protein film packaging materials. Further studies could focus on changing the quality of CNC to make the film more accurately for specific food packaging.

## Figures and Tables

**Figure 1 foods-11-03010-f001:**
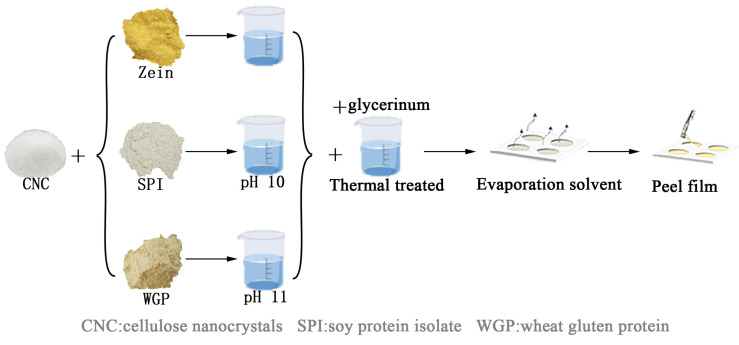
Schematic diagram of the preparation of the plant protein films.

**Figure 2 foods-11-03010-f002:**
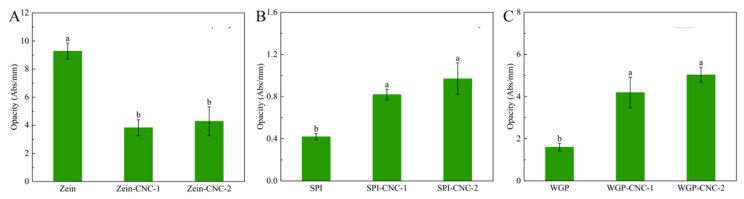
(**A**–**C**) The opacity of Zein/SPI/WGP-based films (Zein: zein; SPI: soy protein isolate; WGP: wheat gluten protein). Different letters (a–c) indicated significant differences (*p* < 0.05).

**Figure 3 foods-11-03010-f003:**
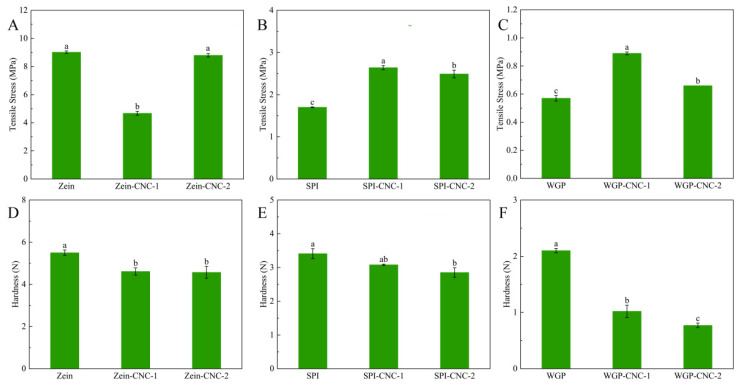
Tensile stress (**A**–**C**) and hardness (**D**–**F**) of the Zein/SPI/WGP-based films. Different letters (a–c) indicated significant differences (*p* < 0.05).

**Figure 4 foods-11-03010-f004:**
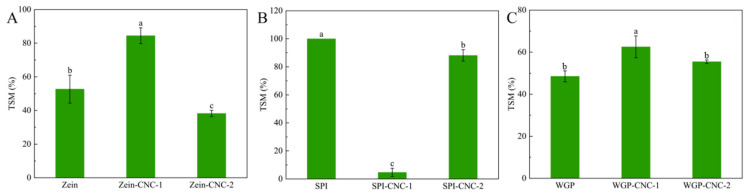
(**A**–**C**) Total soluble matter of the Zein/SPI/WGP-based films. Different letters (a–c) indicated significant differences (*p* < 0.05).

**Figure 5 foods-11-03010-f005:**
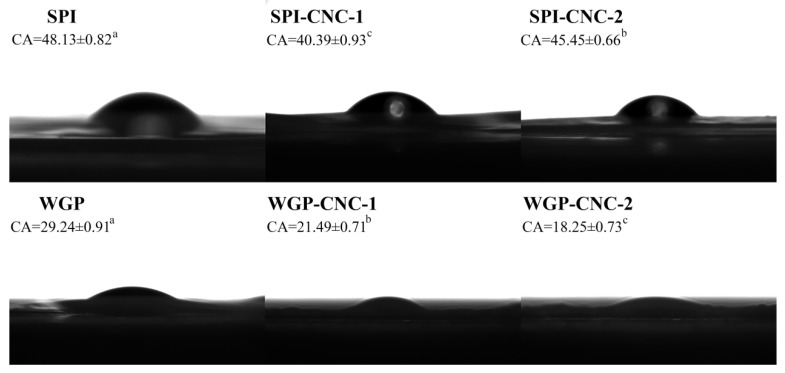
Water contact angle images on the surface of the SPI/WGP-based films. Different letters (a–c) indicated significant differences (*p* < 0.05).

**Figure 7 foods-11-03010-f007:**
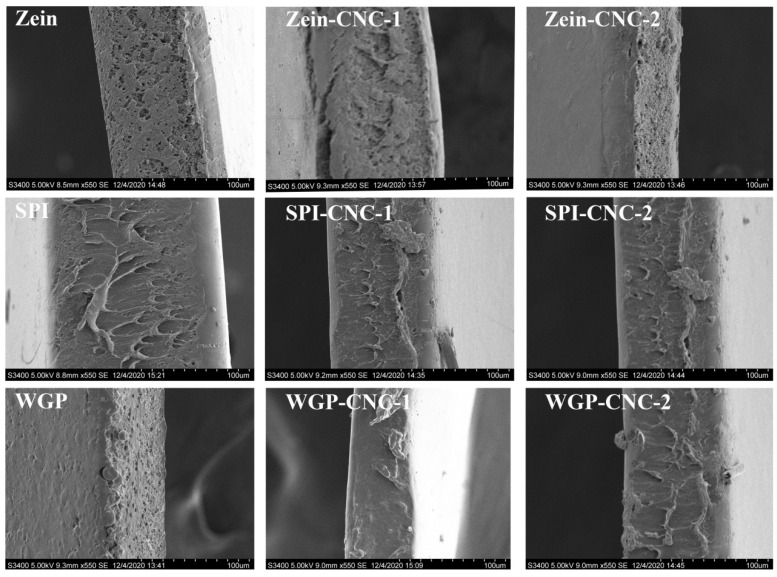
SEM images of cross-section of the Zein/SPI/WGP-based films (Magnifications are 550×).

**Table 1 foods-11-03010-t001:** The thickness and color parameters of Zein-based films with different adding amounts of CNC (CNC-1: 0.25%; CNC-2: 5%). Different letters (a–c) in the same column indicated significant differences (*p* < 0.05).

Film	Thickness (μm)	Color Parameters			
		ΔL*	Δa*	Δb*	ΔE*
Zein	126.67 ± 1.15 ^b^	89.24 ± 0.79 ^b^	−2.03 ± 0.06 ^a^	31.08 ± 0.54 ^a^	33.13 ± 0.44 ^a^
Zein-CNC-1	163.00 ± 1.00 ^a^	91.43 ± 0.11 ^a^	−2.46 ± 0.05 ^b^	29.54 ± 1.02 ^a^	30.87 ± 1.00 ^a^
Zein-CNC-2	86.67 ± 1.16 ^c^	90.68 ± 0.61 ^a^	−2.24 ± 0.20 ^b^	31.69 ± 0.62 ^a^	33.12 ± 0.75 ^a^

**Table 2 foods-11-03010-t002:** The thickness and color parameters of SPI-based films with different adding amounts of CNC (CNC-1: 0.25%; CNC-2: 5%). Different letters (a–c) in the same column indicated significant differences (*p* < 0.05).

Film	Thickness (μm)	Color Parameters			
		ΔL*	Δa*	Δb*	ΔE*
SPI	104.33 ± 0.58 ^a^	95.29 ± 0.39 ^a^	−2.47 ± 0.11 ^a^	9.18 ± 0.35 ^a^	10.71 ± 0.48 ^b^
SPI-CNC-1	103.67 ± 1.16 ^a^	94.84 ± 0.31 ^a^	−3.01 ± 0.31 ^b^	11.20 ± 1.43 ^a^	12.69 ± 1.46 ^a^
SPI-CNC-2	99.33 ± 0.58 ^b^	94.89 ± 0.29 ^a^	−2.92 ± 0.30 ^ab^	10.58 ± 1.35 ^a^	9.73 ± 0.25 ^b^

**Table 3 foods-11-03010-t003:** The thickness and color parameters of WGP-based films with different adding amounts of CNC (CNC-1: 0.25%; CNC-2: 5%). Different letters (a–c) in the same column indicated significant differences (*p* < 0.05).

Film	Thickness (μm)	Color Parameters			
		ΔL*	Δa*	Δb*	ΔE*
WGP	109.00 ± 1.00 ^c^	95.10 ± 0.17 ^a^	−1.83 ± 0.03 ^a^	6.28 ± 0.10 ^b^	8.25 ± 0.14 ^b^
WGP-CNC-1	128.00 ± 0.00 ^b^	93.84 ± 0.50 ^b^	−2.44 ± 0.16 ^b^	8.61 ± 0.84 ^a^	10.88 ± 0.62 ^a^
WGP-CNC-2	154.67 ± 2.08 ^a^	94.68 ± 0.34 ^a^	−2.25 ± 0.01 ^b^	7.83 ± 0.25 ^a^	9.73 ± 0.25 ^c^

**Table 4 foods-11-03010-t004:** The moisture content and water vapor permeability of Zein-based films with different adding amounts of CNC (CNC-1: 0.25%; CNC-2: 5%). Different letters (a–c) in the same column indicated significant differences (*p* < 0.05).

Film	Moisture Content (%)	Water Vapor Permeability (g·mm/(m^2^·d·KPa))
Zein	2.26 ± 0.48 ^a^	14.46 ± 4.35 ^ab^
Zein-CNC-1	1.92 ± 0.42 ^a^	24.88 ± 9.33 ^a^
Zein-CNC-2	1.55 ± 0.34 ^a^	11.57 ± 2.98 ^b^

**Table 5 foods-11-03010-t005:** The moisture content and water vapor permeability of SPI-based films with different adding amounts of CNC (CNC-1: 0.25%; CNC-2: 5%). Different letters (a–c) in the same column indicated significant differences (*p* < 0.05).

Film	Moisture Content (%)	Water Vapor Permeability (g·mm/(m^2^·d·KPa))
SPI	1.76 ± 0.12 ^a^	15.79 ± 3.54 ^a^
SPI-CNC-1	1.17 ± 0.59 ^a^	22.75 ± 3.57 ^a^
SPI-CNC-2	1.63 ± 0.31 ^a^	20.21 ± 3.59 ^a^

**Table 6 foods-11-03010-t006:** The moisture content and water vapor permeability of WGP-based films with different adding amounts of CNC (CNC-1: 0.25%; CNC-2: 5%). Different letters (a–c) in the same column indicated significant differences (*p* < 0.05).

Film	Moisture Content (%)	Water Vapor Permeability (g·mm/(m^2^·d·KPa))
WGP	1.10 ± 0.23 ^a^	20.76 ± 4.00 ^b^
WGP-CNC-1	0.97 ± 0.69 ^a^	39.07 ± 4.88 ^a^
WGP-CNC-2	1.53 ± 0.23 ^a^	20.66 ± 5.32 ^b^

## Data Availability

Data will be made available on request.

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
