# Peer review of "Research on the Properties of Zein, Soy Protein Isolate, and Wheat Gluten Protein-Based Films Containing Cellulose Nanocrystals"

_foods, 2022, doi:10.3390/foods11193010_

Round 1

Reviewer 1 Report

General comments:

The study investigated the effect of CNC on the physical properties, water resistance and mechanical properties of the three protein films (Zein, SPI and WGP). In general, the work is well written, the methodology used is well applied and the results are well discussed and compared to the other authors in the field. However, I believe that some points should be described and justified in more detail.

Specifics comments:

Introduction:

More literatures should be cited to explain the research status and the significance of this study. What are the application fields?

Line 31: “.20×1010 t by 2050” should be Kindly checked and corrected.

Materials and Methods:

Line 146: It should be Moisture content(%)=(m1-m2)/m1×100 and it is better to use subscript.

Line 155: Correct the same as above.

Results and discussion:

Line 185: The author described that the thickness of the film depends on the total amount of non-solvent in film-forming solution, especially the content of solid. So, the comparison of thickness after standardizing should be better because the total amounts of non-solvent in film-forming solutions with and without CNC were different.

Line 212-222: The authors did not provide a single reference, to how their results are related to other studies. The authors have simply explained the results and should provide the reason why these results are obtained.

Line 223-236: The properties of zein, soy protein isolate and wheat gluten protein based films should be compared and analysed.

Line 240: Elongation at break is also an important index of mechanical properties. It is better to supplement the experimental results.

Line 354: How many tests were done? Why there was no standard deviation?

Line 381: Please confirm whether the scale is accurate because the thicknesses in Figure 6 were not consistent with the results of Table 1-3.

How about thermal stability, which is an important property of films?

Reviewer 2 Report

Dear Food editor, I have thoroughly reviewed the manuscript entitled " Research on the properties of zein, soy protein iso- 2 late and wheat gluten protein based films containing cellu- 3 lose nanocrystals" sent to your antioxidant. My decision is minor revisión

Comments for the autor

1.       Line 37-38. Include citations for the following information "Zein is an alcohol soluble protein in maize endosperm, accounting for about 60% of the total zein". Cite: (2020), Food Science and Biotechnology, 29(5), 619-629; (2019), Journal of food science, 84(4), 818-831.

2.       Include the following information in the introduction "Zein is a biodegradable, biocompatible and low toxicity biopolymer, as well, generally recognized as safe by the US Food Drug Administration". Cite: (2022), Journal of Food Engineering, 334, 111153.

3.       Include the following information "Wheat gluten is a by-product of the starch production industry for the production of biofuels and is defined as the gummy mass that results when wheat dough is washed to remove the starch granules"...."Gliadin is a monomeric protein with a molecular weight of 30,000–80,000 Da and is classified in terms of electrophoresis base mobility at a low pH in ω-, α + β-, and γ-gliadins, depending on their molecular weight and amino acid composition, containing intrachain disulfide bonds.Glutenine is a polymeric protein with a molecular weight ranging from 500,000 to several million Da.It is one of the more natural proteins and contains two types of subunits: subunits of low molecular weight (LMW), and subunits of high molecular weight (HMW) containing intrachain and interchain disulfide bonds I recommend the following citation: (2018), Advances in Polymer Technology, 37(6), 2314-2324.

4.       Include a figure of the method of obtaining the films.
